# Dropout Training as Adaptive Regularization

**Stefan Wager**[*], **Sida Wang**[†], and **Percy Liang**[†]
Departments of Statistics[*] and Computer Science[†]
Stanford University, Stanford, CA-94305
swager@stanford.edu,{sidaw, pliang}@cs.stanford.edu

## Abstract

Dropout and other feature noising schemes control overfitting by artificially corrupting the training data. For generalized linear models, dropout performs a form of adaptive regularization. Using this viewpoint, we show that the dropout regularizer is first-order equivalent to an $L_2$ regularizer applied after scaling the features by an estimate of the inverse diagonal Fisher information matrix. We also establish a connection to AdaGrad, an online learning algorithm, and find that a close relative of AdaGrad operates by repeatedly solving linear dropout-regularized problems. By casting dropout as regularization, we develop a natural semi-supervised algorithm that uses unlabeled data to create a better adaptive regularizer. We apply this idea to document classification tasks, and show that it consistently boosts the performance of dropout training, improving on state-of-the-art results on the IMDB reviews dataset.

## 1 Introduction

Dropout training was introduced by Hinton et al. [1] as a way to control overfitting by randomly omitting subsets of features at each iteration of a training procedure.[1] Although dropout has proved to be a very successful technique, the reasons for its success are not yet well understood at a theoretical level.

Dropout training falls into the broader category of learning methods that artificially corrupt training data to stabilize predictions [2, 4, 5, 6, 7]. There is a well-known connection between artificial feature corruption and regularization [8, 9, 10]. For example, Bishop [9] showed that the effect of training with features that have been corrupted with additive Gaussian noise is equivalent to a form of $L_2$-type regularization in the low noise limit. In this paper, we take a step towards understanding how dropout training works by analyzing it as a regularizer. We focus on generalized linear models (GLMs), a class of models for which feature dropout reduces to a form of adaptive model regularization.

Using this framework, we show that dropout training is first-order equivalent to $L_2$-regularization after transforming the input by $\mathrm{diag}(\hat{\mathcal{I}})^{-1/2}$, where $\hat{\mathcal{I}}$ is an estimate of the Fisher information matrix. This transformation effectively makes the level curves of the objective more spherical, and so balances out the regularization applied to different features. In the case of logistic regression, dropout can be interpreted as a form of adaptive $L_2$-regularization that favors rare but useful features.

The problem of learning with rare but useful features is discussed in the context of online learning by Duchi et al. [11], who show that their AdaGrad adaptive descent procedure achieves better regret bounds than regular stochastic gradient descent (SGD) in this setting. Here, we show that AdaGrad

---

S.W. is supported by a B.C. and E.J. Eaves Stanford Graduate Fellowship.

[1]Hinton et al. introduced dropout training in the context of neural networks specifically, and also advocated omitting random hidden layers during training. In this paper, we follow [2, 3] and study feature dropout as a generic training method that can be applied to any learning algorithm.

and dropout training have an intimate connection: Just as SGD progresses by repeatedly solving linearized $L_2$-regularized problems, a close relative of AdaGrad advances by solving linearized dropout-regularized problems.

Our formulation of dropout training as adaptive regularization also leads to a simple semi-supervised learning scheme, where we use unlabeled data to learn a better dropout regularizer. The approach is fully discriminative and does not require fitting a generative model. We apply this idea to several document classification problems, and find that it consistently improves the performance of dropout training. On the benchmark IMDB reviews dataset introduced by [12], dropout logistic regression with a regularizer tuned on unlabeled data outperforms previous state-of-the-art. In follow-up research [13], we extend the results from this paper to more complicated structured prediction, such as multi-class logistic regression and linear chain conditional random fields.

## 2 Artificial Feature Noising as Regularization

We begin by discussing the general connections between feature noising and regularization in generalized linear models (GLMs). We will apply the machinery developed here to dropout training in Section 4.

A GLM defines a conditional distribution over a response $y \in \mathcal{Y}$ given an input feature vector $x \in \mathbb{R}^d$:

$$p_\beta(y \mid x) \stackrel{\text{def}}{=} h(y) \exp\{y\, x \cdot \beta - A(x \cdot \beta)\}, \quad \ell_{x,y}(\beta) \stackrel{\text{def}}{=} -\log p_\beta(y \mid x). \tag{1}$$

Here, $h(y)$ is a quantity independent of $x$ and $\beta$, $A(\cdot)$ is the log-partition function, and $\ell_{x,y}(\beta)$ is the loss function (i.e., the negative log likelihood); Table 1 contains a summary of notation. Common examples of GLMs include linear ($\mathcal{Y} = \mathbb{R}$), logistic ($\mathcal{Y} = \{0, 1\}$), and Poisson ($\mathcal{Y} = \{0, 1, 2, \dots\}$) regression.

Given $n$ training examples $(x_i, y_i)$, the standard maximum likelihood estimate $\hat{\beta} \in \mathbb{R}^d$ minimizes the empirical loss over the training examples:

$$\hat{\beta} \stackrel{\text{def}}{=} \arg\min_{\beta \in \mathbb{R}^d} \sum_{i=1}^{n} \ell_{x_i, y_i}(\beta). \tag{2}$$

With artificial feature noising, we replace the observed feature vectors $x_i$ with noisy versions $\tilde{x}_i = \nu(x_i, \xi_i)$, where $\nu$ is our noising function and $\xi_i$ is an independent random variable. We first create many noisy copies of the dataset, and then average out the auxiliary noise. In this paper, we will consider two types of noise:

- Additive Gaussian noise: $\nu(x_i, \xi_i) = x_i + \xi_i$, where $\xi_i \sim \mathcal{N}(0, \sigma^2 I_{d\times d})$.
- Dropout noise: $\nu(x_i, \xi_i) = x_i \odot \xi_i$, where $\odot$ is the elementwise product of two vectors. Each component of $\xi_i \in \{0, (1 - \delta)^{-1}\}^d$ is an independent draw from a scaled Bernoulli$(1 - \delta)$ random variable. In other words, dropout noise corresponds to setting $\tilde{x}_{ij}$ to 0 with probability $\delta$ and to $x_{ij}/(1 - \delta)$ else.[2]

Integrating over the feature noise gives us a noised maximum likelihood parameter estimate:

$$\hat{\beta} = \arg\min_{\beta \in \mathbb{R}^d} \sum_{i=1}^{n} \mathbb{E}_\xi \left[ \ell_{\tilde{x}_i, y_i}(\beta) \right], \text{ where } \mathbb{E}_\xi \left[ Z \right] \stackrel{\text{def}}{=} \mathbb{E} \left[ Z \mid \{x_i, y_i\} \right] \tag{3}$$

is the expectation taken with respect to the artificial feature noise $\xi = (\xi_1, \dots, \xi_n)$. Similar expressions have been studied by [9, 10].

For GLMs, the noised empirical loss takes on a simpler form:

$$\sum_{i=1}^{n} \mathbb{E}_\xi \left[ \ell_{\tilde{x}_i, y_i}(\beta) \right] = \sum_{i=1}^{n} -\left( y\, x_i \cdot \beta - \mathbb{E}_\xi \left[ A(\tilde{x}_i \cdot \beta) \right] \right) = \sum_{i=1}^{n} \ell_{x_i, y_i}(\beta) + R(\beta). \tag{4}$$

Table 1: Summary of notation.

| $x_i$ | Observed feature vector | $R(\beta)$ | Noising penalty (5) |
|---|---|---|---|
| $\tilde{x}_i$ | Noised feature vector | $R^{\mathrm{q}}(\beta)$ | Quadratic approximation (6) |
| $A(x \cdot \beta)$ | Log-partition function | $\ell(\beta)$ | Negative log-likelihood (loss) |

The first equality holds provided that $\mathbb{E}_\xi[\tilde{x}_i] = x_i$, and the second is true with the following definition:

$$R(\beta) \stackrel{\text{def}}{=} \sum_{i=1}^n \mathbb{E}_\xi \left[ A(\tilde{x}_i \cdot \beta) \right] - A(x_i \cdot \beta). \tag{5}$$

Here, $R(\beta)$ acts as a regularizer that incorporates the effect of artificial feature noising. In GLMs, the log-partition function $A$ must always be convex, and so $R$ is always positive by Jensen's inequality.

The key observation here is that the effect of artificial feature noising reduces to a penalty $R(\beta)$ that does not depend on the labels $\{y_i\}$. Because of this, artificial feature noising penalizes the complexity of a classifier in a way that does not depend on the accuracy of a classifier. Thus, for GLMs, artificial feature noising is a regularization scheme on the model itself that can be compared with other forms of regularization such as ridge ($L_2$) or lasso ($L_1$) penalization. In Section 6, we exploit the label-independence of the noising penalty and use unlabeled data to tune our estimate of $R(\beta)$.

The fact that $R$ does not depend on the labels has another useful consequence that relates to prediction. The natural prediction rule with artificially noised features is to select $\hat{y}$ to minimize expected loss over the added noise: $\hat{y} = \operatorname{argmin}_y \mathbb{E}_\xi[\ell_{\tilde{x}, y}(\hat{\beta})]$. It is common practice, however, not to noise the inputs and just to output classification decisions based on the original feature vector [1, 3, 14]: $\hat{y} = \operatorname{argmin}_y \ell_{x, y}(\hat{\beta})$. It is easy to verify that these expressions are in general not equivalent, but they are equivalent when the effect of feature noising reduces to a label-independent penalty on the likelihood. Thus, the common practice of predicting with clean features is formally justified for GLMs.

## 2.1 A Quadratic Approximation to the Noising Penalty

Although the noising penalty $R$ yields an explicit regularizer that does not depend on the labels $\{y_i\}$, the form of $R$ can be difficult to interpret. To gain more insight, we will work with a quadratic approximation of the type used by [9, 10]. By taking a second-order Taylor expansion of $A$ around $x \cdot \beta$, we get that $\mathbb{E}_\xi \left[ A(\tilde{x} \cdot \beta) \right] - A(x \cdot \beta) \approx \frac{1}{2} A''(x \cdot \beta) \operatorname{Var}_\xi \left[ \tilde{x} \cdot \beta \right]$. Here the first-order term $\mathbb{E}_\xi \left[ A'(x \cdot \beta)(\tilde{x} - x) \right]$ vanishes because $\mathbb{E}_\xi[\tilde{x}] = x$. Applying this quadratic approximation to (5) yields the following *quadratic noising regularizer*, which will play a pivotal role in the rest of the paper:

$$R^{\mathrm{q}}(\beta) \stackrel{\text{def}}{=} \frac{1}{2} \sum_{i=1}^n A''(x_i \cdot \beta) \operatorname{Var}_\xi \left[ \tilde{x}_i \cdot \beta \right]. \tag{6}$$

This regularizer penalizes two types of variance over the training examples: (i) $A''(x_i \cdot \beta)$, which corresponds to the variance of the response $y_i$ in the GLM, and (ii) $\operatorname{Var}_\xi[\tilde{x}_i \cdot \beta]$, the variance of the estimated GLM parameter due to noising.[3]

**Accuracy of approximation** Figure 1a compares the noising penalties $R$ and $R^{\mathrm{q}}$ for logistic regression in the case that $\tilde{x} \cdot \beta$ is Gaussian;[4] we vary the mean parameter $p \stackrel{\text{def}}{=} (1 + e^{-x \cdot \beta})^{-1}$ and the noise level $\sigma$. We see that $R^{\mathrm{q}}$ is generally very accurate, although it tends to overestimate the true penalty for $p \approx 0.5$ and tends to underestimate it for very confident predictions. We give a graphical explanation for this phenomenon in the Appendix (Figure A.1).

The quadratic approximation also appears to hold up on real datasets. In Figure 1b, we compare the evolution during training of both $R$ and $R^{\mathrm{q}}$ on the 20 newsgroups `alt.atheism` vs

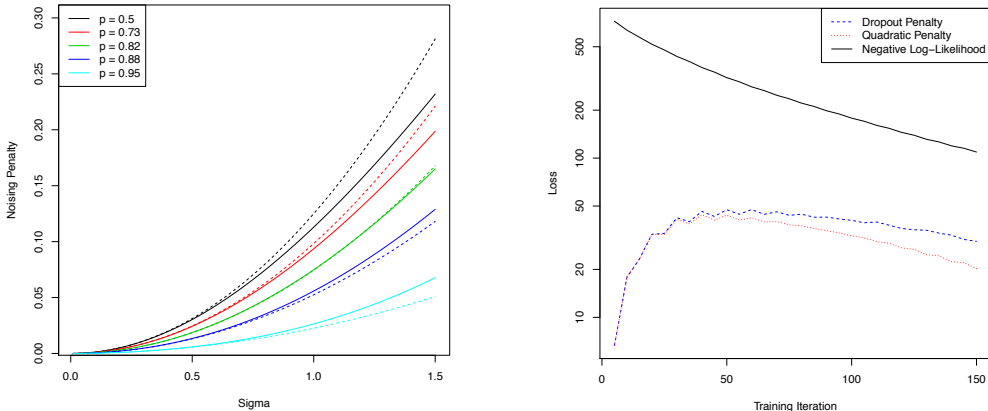

(a) Comparison of noising penalties $R$ and $R^q$ for logistic regression with Gaussian perturbations, i.e., $(\tilde{x} - x) \cdot \beta \sim \mathcal{N}(0, \sigma^2)$. The solid line indicates the true penalty and the dashed one is our quadratic approximation thereof; $p = (1 + e^{-x \cdot \beta})^{-1}$ is the mean parameter for the logistic model.

(b) Comparing the evolution of the exact dropout penalty $R$ and our quadratic approximation $R^q$ for logistic regression on the `AthR` classification task in [15] with 22K features and $n = 1000$ examples. The horizontal axis is the number of quasi-Newton steps taken while training with exact dropout.

Figure 1: Validating the quadratic approximation.

`soc.religion.christian` classification task described in [15]. We see that the quadratic approximation is accurate most of the way through the learning procedure, only deteriorating slightly as the model converges to highly confident predictions.

In practice, we have found that fitting logistic regression with the quadratic surrogate $R^q$ gives similar results to actual dropout-regularized logistic regression. We use this technique for our experiments in Section 6.

## 3 Regularization based on Additive Noise

Having established the general quadratic noising regularizer $R^q$, we now turn to studying the effects of $R^q$ for various likelihoods (linear and logistic regression) and noising models (additive and dropout). In this section, we warm up with additive noise; in Section 4 we turn to our main target of interest, namely dropout noise.

**Linear regression**  Suppose $\tilde{x} = x + \varepsilon$ is generated by by adding noise with $\mathrm{Var}[\varepsilon] = \sigma^2 I_{d \times d}$ to the original feature vector $x$. Note that $\mathrm{Var}_\xi[\tilde{x} \cdot \beta] = \sigma^2 \|\beta\|_2^2$, and in the case of linear regression $A(z) = \frac{1}{2}z^2$, so $A''(z) = 1$. Applying these facts to (6) yields a simplified form for the quadratic noising penalty:

$$R^q(\beta) = \frac{1}{2}\sigma^2 n \|\beta\|_2^2. \tag{7}$$

Thus, we recover the well-known result that linear regression with additive feature noising is equivalent to ridge regression [2, 9]. Note that, with linear regression, the quadratic approximation $R^q$ is exact and so the correspondence with $L_2$-regularization is also exact.

**Logistic regression**  The situation gets more interesting when we move beyond linear regression. For logistic regression, $A''(x_i \cdot \beta) = p_i(1 - p_i)$ where $p_i = (1 + \exp(-x_i \cdot \beta))^{-1}$ is the predicted probability of $y_i = 1$. The quadratic noising penalty is then

$$R^q(\beta) = \frac{1}{2}\sigma^2 \|\beta\|_2^2 \sum_{i=1}^{n} p_i(1 - p_i). \tag{8}$$

In other words, the noising penalty now simultaneously encourages parsimonious modeling as before (by encouraging $\|\beta\|_2^2$ to be small) as well as confident predictions (by encouraging the $p_i$'s to move away from $\frac{1}{2}$).

Table 2: Form of the different regularization schemes. These expressions assume that the design matrix has been normalized, i.e., that $\sum_i x_{ij}^2 = 1$ for all $j$. The $p_i = (1 + e^{-x_i \cdot \beta})^{-1}$ are mean parameters for the logistic model.

| | Linear Regression | Logistic Regression | GLM |
|---|---|---|---|
| $L_2$-penalization | $\|\beta\|_2^2$ | $\|\beta\|_2^2$ | $\|\beta\|_2^2$ |
| Additive Noising | $\|\beta\|_2^2$ | $\|\beta\|_2^2 \sum_i p_i(1 - p_i)$ | $\|\beta\|_2^2 \operatorname{tr}(V(\beta))$ |
| Dropout Training | $\|\beta\|_2^2$ | $\sum_{i,j} p_i(1 - p_i) x_{ij}^2 \beta_j^2$ | $\beta^\top \operatorname{diag}(X^\top V(\beta)X)\beta$ |

## 4  Regularization based on Dropout Noise

Recall that dropout training corresponds to applying dropout noise to training examples, where the noised features $\tilde{x}_i$ are obtained by setting $\tilde{x}_{ij}$ to 0 with some "dropout probability" $\delta$ and to $x_{ij}/(1 - \delta)$ with probability $(1 - \delta)$, independently for each coordinate $j$ of the feature vector. We can check that:

$$\operatorname{Var}_\xi [\tilde{x}_i \cdot \beta] = \frac{1}{2} \frac{\delta}{1 - \delta} \sum_{j=1}^d x_{ij}^2 \beta_j^2, \tag{9}$$

and so the *quadratic dropout penalty* is

$$R^{\mathrm{q}}(\beta) = \frac{1}{2} \frac{\delta}{1 - \delta} \sum_{i=1}^n A''(x_i \cdot \beta) \sum_{j=1}^d x_{ij}^2 \beta_j^2. \tag{10}$$

Letting $X \in \mathbb{R}^{n \times d}$ be the design matrix with rows $x_i$ and $V(\beta) \in \mathbb{R}^{n \times n}$ be a diagonal matrix with entries $A''(x_i \cdot \beta)$, we can re-write this penalty as

$$R^{\mathrm{q}}(\beta) = \frac{1}{2} \frac{\delta}{1 - \delta} \beta^\top \operatorname{diag}(X^\top V(\beta)X)\beta. \tag{11}$$

Let $\beta^*$ be the maximum likelihood estimate given infinite data. When computed at $\beta^*$, the matrix $\frac{1}{n} X^\top V(\beta^*)X = \frac{1}{n} \sum_{i=1}^n \nabla^2 \ell_{x_i, y_i}(\beta^*)$ is an estimate of the Fisher information matrix $\mathcal{I}$. Thus, dropout can be seen as an attempt to apply an $L_2$ penalty after normalizing the feature vector by $\operatorname{diag}(\mathcal{I})^{-1/2}$. The Fisher information is linked to the shape of the level surfaces of $\ell(\beta)$ around $\beta^*$. If $\mathcal{I}$ were a multiple of the identity matrix, then these level surfaces would be perfectly spherical around $\beta^*$. Dropout, by normalizing the problem by $\operatorname{diag}(\mathcal{I})^{-1/2}$, ensures that while the level surfaces of $\ell(\beta)$ may not be spherical, the $L_2$-penalty is applied in a basis where the features have been balanced out. We give a graphical illustration of this phenomenon in Figure A.2.

**Linear Regression**  For linear regression, $V$ is the identity matrix, so the dropout objective is equivalent to a form of ridge regression where each column of the design matrix is normalized before applying the $L_2$ penalty.[5] This connection has been noted previously by [3].

**Logistic Regression**  The form of dropout penalties becomes much more intriguing once we move beyond the realm of linear regression. The case of logistic regression is particularly interesting. Here, we can write the quadratic dropout penalty from (10) as

$$R^{\mathrm{q}}(\beta) = \frac{1}{2} \frac{\delta}{1 - \delta} \sum_{i=1}^n \sum_{j=1}^d p_i(1 - p_i) x_{ij}^2 \beta_j^2. \tag{12}$$

Thus, just like additive noising, dropout generally gives an advantage to confident predictions and small $\beta$. However, unlike all the other methods considered so far, dropout may allow for some large $p_i(1 - p_i)$ and some large $\beta_j^2$, provided that the corresponding cross-term $x_{ij}^2$ is small.

Our analysis shows that dropout regularization should be better than $L_2$-regularization for learning weights for features that are rare (i.e., often 0) but highly discriminative, because dropout effectively does not penalize $\beta_j$ over observations for which $x_{ij} = 0$. Thus, in order for a feature to earn a large $\beta_j^2$, it suffices for it to contribute to a confident prediction with small $p_i(1 - p_i)$ each time that it is active.[6] Dropout training has been empirically found to perform well on tasks such as document

Table 3: Accuracy of $L_2$ and dropout regularized logistic regression on a simulated example. The first row indicates results over test examples where some of the rare useful features are active (i.e., where there is some signal that can be exploited), while the second row indicates accuracy over the full test set. These results are averaged over 100 simulation runs, with 75 training examples in each. All tuning parameters were set to optimal values. The sampling error on all reported values is within $\pm 0.01$.

| Accuracy | $L_2$-regularization | Dropout training |
|---|---|---|
| Active Instances | 0.66 | **0.73** |
| All Instances | 0.53 | **0.55** |

classification where rare but discriminative features are prevalent [3]. Our result suggests that this is no mere coincidence.

We summarize the relationship between $L_2$-penalization, additive noising and dropout in Table 2. Additive noising introduces a product-form penalty depending on both $\beta$ and $A''$. However, the full potential of artificial feature noising only emerges with dropout, which allows the penalty terms due to $\beta$ and $A''$ to interact in a non-trivial way through the design matrix $X$ (except for linear regression, in which all the noising schemes we consider collapse to ridge regression).

### 4.1 A Simulation Example

The above discussion suggests that dropout logistic regression should perform well with rare but useful features. To test this intuition empirically, we designed a simulation study where all the signal is grouped in 50 rare features, each of which is active only 4% of the time. We then added 1000 nuisance features that are always active to the design matrix, for a total of $d = 1050$ features. To make sure that our experiment was picking up the effect of dropout training specifically and not just normalization of $X$, we ensured that the columns of $X$ were normalized in expectation.

The dropout penalty for logistic regression can be written as a matrix product

$$R^{\mathsf{q}}(\beta) = \frac{1}{2} \frac{\delta}{1-\delta} \left( \cdots \quad p_i(1-p_i) \quad \cdots \right) \begin{pmatrix} & \cdots & \\ \cdots & x_{ij}^2 & \cdots \\ & \cdots & \end{pmatrix} \begin{pmatrix} \cdots \\ \beta_j^2 \\ \cdots \end{pmatrix}. \tag{13}$$

We designed the simulation study in such a way that, at the optimal $\beta$, the dropout penalty should have structure

$$\left( \underset{\substack{\text{Small} \\ \text{(confident prediction)}}}{} \middle| \underset{\substack{\textbf{Big} \\ \text{(weak prediction)}}}{} \right) \left( \begin{array}{c|c} \cdots & \cdots \\ \hline \mathbf{0} & \cdots \end{array} \right) \left( \begin{array}{c} \underset{\substack{\textbf{Big} \\ \text{(useful feature)}}}{} \\ - \\ \underset{\substack{\text{Small} \\ \text{(nuisance feature)}}}{} \end{array} \right). \tag{14}$$

A dropout penalty with such a structure should be small. Although there are some uncertain predictions with large $p_i(1-p_i)$ and some big weights $\beta_j^2$, these terms cannot interact because the corresponding terms $x_{ij}^2$ are all 0 (these are examples without any of the rare discriminative features and thus have no signal). Meanwhile, $L_2$ penalization has no natural way of penalizing some $\beta_j$ more and others less. Our simulation results, given in Table 3, confirm that dropout training outperforms $L_2$-regularization here as expected. See Appendix A.1 for details.

## 5 Dropout Regularization in Online Learning

There is a well-known connection between $L_2$-regularization and stochastic gradient descent (SGD). In SGD, the weight vector $\hat{\beta}$ is updated with $\hat{\beta}_{t+1} = \hat{\beta}_t - \eta_t g_t$, where $g_t = \nabla \ell_{x_t, y_t}(\hat{\beta}_t)$ is the gradient of the loss due to the $t$-th training example. We can also write this update as a linear $L_2$-penalized problem

$$\hat{\beta}_{t+1} = \operatorname{argmin}_\beta \left\{ \ell_{x_t, y_t}(\hat{\beta}_t) + g_t \cdot (\beta - \hat{\beta}_t) + \frac{1}{2\eta_t} \|\beta - \hat{\beta}_t\|_2^2 \right\}, \tag{15}$$

where the first two terms form a linear approximation to the loss and the third term is an $L_2$-regularizer. Thus, SGD progresses by repeatedly solving linearized $L_2$-regularized problems.

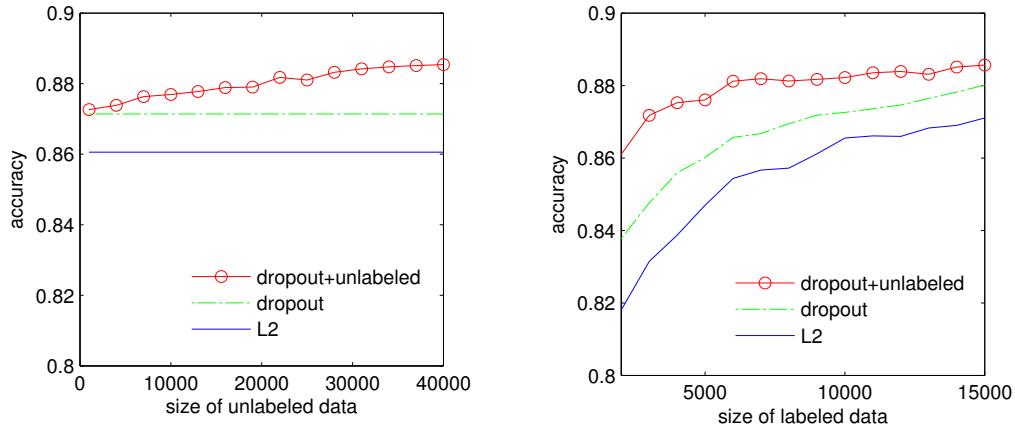

Figure 2: Test set accuracy on the IMDB dataset [12] with unigram features. Left: 10000 labeled training examples, and up to 40000 unlabeled examples. Right: 3000-15000 labeled training examples, and 25000 unlabeled examples. The unlabeled data is discounted by a factor $\alpha = 0.4$.

As discussed by Duchi et al. [11], a problem with classic SGD is that it can be slow at learning weights corresponding to rare but highly discriminative features. This problem can be alleviated by running a modified form of SGD with $\hat{\beta}_{t+1} = \hat{\beta}_t - \eta \, A_t^{-1} g_t$, where the transformation $A_t$ is also learned online; this leads to the AdaGrad family of stochastic descent rules. Duchi et al. use $A_t = \mathrm{diag}(G_t)^{1/2}$ where $G_t = \sum_{i=1}^t g_i g_i^\top$ and show that this choice achieves desirable regret bounds in the presence of rare but useful features. At least superficially, AdaGrad and dropout seem to have similar goals: For logistic regression, they can both be understood as adaptive alternatives to methods based on $L_2$-regularization that favor learning rare, useful features. As it turns out, they have a deeper connection.

The natural way to incorporate dropout regularization into SGD is to replace the penalty term $\|\beta - \hat{\beta}\|_2^2/2\eta$ in (15) with the dropout regularizer, giving us an update rule

$$\hat{\beta}_{t+1} = \mathrm{argmin}_\beta \left\{ \ell_{x_t, y_t}(\hat{\beta}_t) + g_t \cdot (\beta - \hat{\beta}_t) + R^q(\beta - \hat{\beta}_t; \hat{\beta}_t) \right\} \qquad (16)$$

where, $R^q(\cdot; \hat{\beta}_t)$ is the quadratic noising regularizer centered at $\hat{\beta}_t$:[7]

$$R^q(\beta - \hat{\beta}_t; \hat{\beta}_t) = \frac{1}{2}(\beta - \hat{\beta}_t)^\top \mathrm{diag}(H_t) \, (\beta - \hat{\beta}_t), \text{where } H_t = \sum_{i=1}^t \nabla^2 \ell_{x_i, y_i}(\hat{\beta}_t). \qquad (17)$$

This implies that dropout descent is first-order equivalent to an adaptive SGD procedure with $A_t = \mathrm{diag}(H_t)$. To see the connection between AdaGrad and this dropout-based online procedure, recall that for GLMs both of the expressions

$$\mathbb{E}_{\beta^*}\left[\nabla^2 \ell_{x, y}(\beta^*)\right] = \mathbb{E}_{\beta^*}\left[\nabla \ell_{x, y}(\beta^*) \nabla \ell_{x, y}(\beta^*)^\top\right] \qquad (18)$$

are equal to the Fisher information $\mathcal{I}$ [17]. In other words, as $\hat{\beta}_t$ converges to $\beta^*$, $G_t$ and $H_t$ are both consistent estimates of the Fisher information. Thus, by using dropout instead of $L_2$-regularization to solve linearized problems in online learning, we end up with an AdaGrad-like algorithm.

Of course, the connection between AdaGrad and dropout is not perfect. In particular, AdaGrad allows for a more aggressive learning rate by using $A_t = \mathrm{diag}(G_t)^{-1/2}$ instead of $\mathrm{diag}(G_t)^{-1}$. But, at a high level, AdaGrad and dropout appear to both be aiming for the same goal: scaling the features by the Fisher information to make the level-curves of the objective more circular. In contrast, $L_2$-regularization makes no attempt to sphere the level curves, and AROW [18]—another popular adaptive method for online learning—only attempts to normalize the effective feature matrix but does not consider the sensitivity of the loss to changes in the model weights. In the case of logistic regression, AROW also favors learning rare features, but unlike dropout and AdaGrad does not privilege confident predictions.

Table 4: Performance of semi-supervised dropout training for document classification.

(a) Test accuracy with and without unlabeled data on different datasets. Each dataset is split into 3 parts of equal sizes: train, unlabeled, and test. Log. Reg.: logistic regression with $L_2$ regularization; Dropout: dropout trained with quadratic surrogate; +Unlabeled: using unlabeled data.

| Datasets | Log. Reg. | Dropout | +Unlabeled |
|---|---|---|---|
| Subj | 88.96 | 90.85 | **91.48** |
| RT | 73.49 | 75.18 | **76.56** |
| IMDB-2k | 80.63 | **81.23** | 80.33 |
| XGraph | 83.10 | 84.64 | **85.41** |
| BbCrypt | 97.28 | 98.49 | **99.24** |
| IMDB | 87.14 | 88.70 | **89.21** |

(b) Test accuracy on the IMDB dataset [12]. Labeled: using just labeled data from each paper/method, +Unlabeled: use additional unlabeled data. Drop: dropout with $R^q$, MNB: multionomial naive Bayes with semi-supervised frequency estimate from [19],[8]-Uni: unigram features, -Bi: bigram features.

| Methods | Labeled | +Unlabeled |
|---|---|---|
| MNB-Uni [19] | 83.62 | 84.13 |
| MNB-Bi [19] | 86.63 | 86.98 |
| Vect.Sent[12] | 88.33 | 88.89 |
| NBSVM[15]-Bi | 91.22 | – |
| Drop-Uni | 87.78 | 89.52 |
| Drop-Bi | 91.31 | **91.98** |

# 6 Semi-Supervised Dropout Training

Recall that the regularizer $R(\beta)$ in (5) is independent of the labels $\{y_i\}$. As a result, we can use additional unlabeled training examples to estimate it more accurately. Suppose we have an unlabeled dataset $\{z_i\}$ of size $m$, and let $\alpha \in (0,1]$ be a discount factor for the unlabeled data. Then we can define a semi-supervised penalty estimate

$$R_*(\beta) \stackrel{\text{def}}{=} \frac{n}{n + \alpha m}\Big( R(\beta) + \alpha\, R_{\text{Unlabeled}}(\beta)\Big), \qquad (19)$$

where $R(\beta)$ is the original penalty estimate and $R_{\text{Unlabeled}}(\beta) = \sum_i \mathbb{E}_\xi[A(z_i \cdot \beta)] - A(z_i \cdot \beta)$ is computed using (5) over the unlabeled examples $z_i$. We select the discount parameter by cross-validation; empirically, $\alpha \in [0.1, 0.4]$ works well. For convenience, we optimize the quadratic surrogate $R_*^q$ instead of $R_*$. Another practical option would be to use the Gaussian approximation from [3] for estimating $R_*(\beta)$.

Most approaches to semi-supervised learning either rely on using a generative model [19, 20, 21, 22, 23] or various assumptions on the relationship between the predictor and the marginal distribution over inputs. Our semi-supervised approach is based on a different intuition: we'd like to set weights to make confident predictions on unlabeled data as well as the labeled data, an intuition shared by entropy regularization [24] and transductive SVMs [25].

**Experiments** We apply this semi-supervised technique to text classification. Results on several datasets described in [15] are shown in Table 4a; Figure 2 illustrates how the use of unlabeled data improves the performance of our classifier on a single dataset. Overall, we see that using unlabeled data to learn a better regularizer $R_*(\beta)$ consistently improves the performance of dropout training.

Table 4b shows our results on the IMDB dataset of [12]. The dataset contains 50,000 unlabeled examples in addition to the labeled train and test sets of size 25,000 each. Whereas the train and test examples are either positive or negative, the unlabeled examples contain neutral reviews as well. We train a dropout-regularized logistic regression classifier on unigram/bigram features, and use the unlabeled data to tune our regularizer. Our method benefits from unlabeled data even in the presence of a large amount of labeled data, and achieves state-of-the-art accuracy on this dataset.

# 7 Conclusion

We analyzed dropout training as a form of adaptive regularization. This framework enabled us to uncover close connections between dropout training, adaptively balanced $L_2$-regularization, and AdaGrad; and led to a simple yet effective method for semi-supervised training. There seem to be multiple opportunities for digging deeper into the connection between dropout training and adaptive regularization. In particular, it would be interesting to see whether the dropout regularizer takes on a tractable and/or interpretable form in neural networks, and whether similar semi-supervised schemes could be used to improve on the results presented in [1].

## Footnotes

[2]Artificial noise of the form $x_i \odot \xi_i$ is also called blankout noise. For GLMs, blankout noise is equivalent to dropout noise as defined by [1].

[3] Although $R^{\mathrm{q}}$ is not convex, we were still able (using an L-BFGS algorithm) to train logistic regression with $R^{\mathrm{q}}$ as a surrogate for the dropout regularizer without running into any major issues with local optima.

[4] This assumption holds a priori for additive Gaussian noise, and can be reasonable for dropout by the central limit theorem.

[5]Normalizing the columns of the design matrix before performing penalized regression is standard practice, and is implemented by default in software like `glmnet` for R [16].

[6]To be precise, dropout does not reward all rare but discriminative features. Rather, dropout rewards those features that are rare and positively co-adapted with other features in a way that enables the model to make confident predictions whenever the feature of interest is active.

[7]This expression is equivalent to (11) except that we used $\hat{\beta}_t$ and not $\beta - \hat{\beta}_t$ to compute $H_t$.

[8]Our implementation of semi-supervised MNB. MNB with EM [20] failed to give an improvement.

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
