[Supplementary Material · appendix.pdf]

# A  Appendix

Figure A.1: Quadratic approximations to the logistic loss. We see that the red curve, namely the quadratic approximation taken at $\eta = 0$, $p = 1/(1 + e^{\eta}) = 0.5$ is always above the actual loss curve. Meanwhile, quadratic approximations taken at the more extreme locations of $p = 0.05$ and $p = 0.95$ undershoot the true loss over a large range. Note that the curvature of the loss is symmetric in the natural parameter $\eta$ and so the performance of the quadratic approximation is equivalent at $p$ and $1 - p$ for all $p \in (0, 1)$.

## A.1  Description of Simulation Study

Section 4.1 gives the motivation for and a high-level description of our simulation study. Here, we give a detailed description of the study.

**Generating features.**   Our simulation has 1050 features. The first 50 discriminative features form 5 groups of 10; the last 1000 features are nuisance terms. Each $x_i$ was independently generated as follows:

1. Pick a group number $g \in 1, ..., 25$, and a sign $sgn = \pm 1$.
2. If $g \leq 5$, draw the entries of $x_i$ with index between $10\,(g - 1) + 1$ and $10\,(g - 1) + 10$ uniformly from $sgn \cdot \mathrm{Exp}(C)$, where $C$ is selected such that $\mathbb{E}[(x_i)_j^2] = 1$ for all $j$. Set all the other discriminative features to 0. If $g > 5$, set all the discriminative features to 0.
3. Draw the last 1000 entries of $x_i$ independently from $\mathcal{N}(0, 1)$.

Notice that this procedure guarantees that the columns of $X$ all have the same expected second moments.

**Generating labels.**   Given an $x_i$, we generate $y_i$ from the Bernoulli distribution with parameter $\sigma(x_i \cdot \beta)$, where the first 50 coordinates of $\beta$ are 0.057 and the remaining 1000 coordinates are 0. The value 0.057 was selected to make the average value of $|x_i \cdot \beta|$ in the presence of signal be 2.

**Training.**   For each simulation run, we generated a training set of size $n = 75$. For this purpose, we cycled over the group number $g$ deterministically. The penalization parameters were set to roughly optimal values. For dropout, we used $\delta = 0.9$ while from $L_2$-penalization we used $\lambda = 32$.

Figure A.2: Comparison of two $L_2$ regularizers. In both cases, the black solid ellipses are level surfaces of the likelihood and the blue dashed curves are level surfaces of the regularizer; the optimum of the regularized objective is denoted by OPT. The left panel shows a classic spherical $L_2$ regulizer $\|\beta\|_2^2$, whereas the right panel has an $L_2$ regularizer $\beta^\top \operatorname{diag}(\mathcal{I})\beta$ that has been adapted to the shape of the likelihood ($\mathcal{I}$ is the Fisher information matrix). The second regularizer is still aligned with the axes, but the relative importance of each axis is now scaled using the curvature of the likelihood function. As argued in (11), dropout training is comparable to the setup depicted in the right panel.