[Reviews · NeurIPS 2013]

Submitted by Assigned_Reviewer_4

Update: I have read the rebuttal and my opinion remains unchanged; I still think it's a very nice paper.

Summary of the paper:
In this paper, dropout training is analyzed in terms of it's effect as a regularizer. First the authors discuss Gaussian additive noise and dropout noise in the context of regularizing generalized linear models. They decompose the expected loss into two terms: a loss function over the labels, and a label-independent regularizer based on an expectation over the noise distribution. Using this, they approximate the noise-based regularizer via a second-order Taylor expansion in order to yield closed form regularizers. In the case of linear regression with normalized features, they recover the standard L2 penalty, while for logistic regression they recover novel penalty functions. They reason and demonstrate that the effect of dropout on logistic regression is to penalize large weights for commonly activated features, while allowing large weights for rare but highly discriminative features. They then cast dropout as a form of adaptive SGD in the same vein as AdaGrad. Finally, the authors exploit the fact that the effective dropout penalty function doesn't depend on the labels, and use this to perform semi-supervised learning.

Quality:
I believe this paper is of excellent quality. It has a clear motivation and a careful analysis that is backed up with empirical evidence. I particularly like the non-obvious connection to AdaGrad, as well as the semi-supervised extension. Perhaps the only empirical evaluation that is missing is a speed comparison between the quadratic, deterministic approximation and the stochastic version (similar to Wang and Manning).

Clarity:
This paper is quite clear, and the progresses through the various ideas in a nice cohesive fashion. It would have been nice if some of the more involved derivations not included in the paper had been included in the supplementary material. It's also not clear to me how they recover the exact penalty for the comparison in section 2.1. Did they use a Monte Carlo estimate for the exact penalty?

Originality:
This paper makes several original contributions, namely the deterministic penalty for dropout training of GLMs, an analysis of the effect of dropout on feature weighting, the connection with AdaGrad, and the semi-supervised extension.

Significance:
This paper makes a significant contribution to our understanding of the effect of noise-based regularization in supervised learning. I have no doubt that the results in this paper will be built upon in the future, especially the semi-supervised version.
Summary: This paper was a pleasure to read. It had interesting analysis, novel connections to previous work, interesting extensions, and a relatively thorough empirical evaluation.

Submitted by Assigned_Reviewer_5

This paper describes how the dropout technique (and additive Gaussian
noise) can be seen as a form of adaptive regularization. Dropout has
been shown recently to provide significant performance improvements in
deep learning architectures. However, theoretical explanations
provided so far of why it works so well have not been
conclusive. While this paper is only restricted to some forms of
generalized linear models (which are simpler architectures than
typical deep learning architectures), dropout is shown to be
equivalent to very intuitive adaptive regularization, especially
useful when rare but discriminative features are prevalent. To my
knowledge, this is the first time that such a clear explanation of
dropout is provided. Moreover, a connection is also made with
AdaGrad. A semi-supervised learning algorithm is derived from this
connection, and empirical results show improvement over logistic
regression and the supervised version of the dropout regularizer.

Very interesting and well written paper. I was impressed by the
clarity of the derivations (culminating with Eq. 12), and I'm eager to
see if a similar analysis holds for deep architectures.

One suggestion about the content of this version of the paper (or
eventual subsequent journal versions): Section 4.1 in itself is not
very useful. Although it is interesting, it takes too much space in
the paper for what it's worth. It is always possible to generate
synthetic datasets to advantage any model. Instead, more space could
be devoted to describe more clearly and intuitively the relationship
between the regularizer (Eq. 11) and the Fischer information. A
graphical depiction of the spherical properties of the Fischer
normalization described in lines 243-246 could be useful in that
sense.

Minor comments:

Line 263: x_{ij} should read x^2_{ij}

Provide a reference for Eq. 15.

Line 355: The penalty term is different from Eq. 15.

I have taken into account the authors rebuttal in my final review.
Summary: Very interesting connection between dropout and regularization in generalized linear models. Nice application to semi-supervised learning.

Submitted by Assigned_Reviewer_6

*Summary of the paper:

This paper studies "dropout training" in the framework of feature noising as a regularization.

It is know that adding an additive gaussian noise to the feature is equivalent to an l_2 regularization in a least square problem (Bishop).

This paper studies multiplicative Bernoulli feature noising, in a shallow learning architecture, with a general loss function and shows that it has the effect of adapting the geometry through an "l_2 regularizer " that rescales the feature (beta^{\top} D(beta,X) beta).

The Matrix D(beta,X) is a estimate of the inverse diagonal fisher information.
It is worth noting that D does not depend on the labels. The equivalent regularizer of dropout is non convex in general.

A connexion to AdaGrad in online learning introduced in Duchi et al is made, as both approaches reduces to adapting the geometry towards rare but predictive features.

The Matrix D(beta,X) could be estimated using unlabeled data, authors devise a semi supervised variant of the equivalent dropout regularizer.


Comments:

- Dropout as originally introduced by hinton et al considers multiplicative Bernoulli noising for deep learning architectures (neural nets).
An average neural net is obtained, by training independently many neural nets where some hidden layers were dropped out at random.
Hinton's dropout seems to be close to ensemble methods (boosting, bagging etc.), and a different analysis is required to understand this paradigm even with one layer.

Referring to Hinton's dropout in the introduction is a bit misleading as there is no models averaging in the current paper.
It would be more precise to mention that this work consider another form of dropout, where we learn one model robust to Bernoulli multiplicative perturbation.


- The regularizer is not an l_2 regularizer as D depends on beta also in a non linear way through the hessian of the likelihood. one should be careful with this appellation.

- It is interesting to derive the approximate equivalent regularizer, to understand the effect of dropout. The non convexity of the regularizer poses computational issues. Authors mention in a footnote that they use lbfgs, more explanation and discussion are needed to understand how to avoid local minimas.
The non convexity of the equivalent regularizer is not discussed in section 6.

- The parameter (delta/1-delta) corresponds to a regularization parameter. any insight on how to choose delta? according to Hinton et al , delta=1/2 seems to have a good performance.

- The log partition function is sometimes hard to get in closed from any insight on how to go around that?


Authors answered promptly questions raised by the reviewer.
Summary: The paper is well written, and is a step towards understanding the effect of dropout.
It is though analyzing a different problem than the original dropout as introduced by Hinton et al as an ensemble method: averaging many randomly sampled models, this should be made clear.
This paper shows how to learn a model robust in average against Bernoulli multiplicative noise. while Hinton's dropout shows how to learn an average model of randomly sampled models with dropout.




Author Feedback

Author rebuttal: Thank you for your thoughtful feedback and helpful suggestions. We look forward to revising our paper in light of your comments. Many of the referee comments, especially regarding convexity, highlight interesting directions for new research.

Below are some responses to individual reviewer comments.

Assigned_Reviewer_4:

> Perhaps the only empirical evaluation that is missing is a speed comparison between the quadratic, deterministic approximation and the stochastic version (similar to Wang and Manning).

We did run these experiments. In the case of two class logistic regression, our quadratic approximation behaves very similarly to the Gaussian approximation of Wang and Manning, both in terms of speed and accuracy. The advantage of our second-order expansion is that it generalizes naturally to a semi-supervised setup (as emphasized in this paper). In follow-up research, we have also applied our second-order method to more complicated forms of structured prediction such as conditional random fields.

> It would have been nice if some of the more involved derivations not included in the paper had been included in the supplementary material.

We can add some more detailed derivations to our next draft.

> It's also not clear to me how they recover the exact penalty for the comparison in section 2.1. Did they use a Monte Carlo estimate for the exact penalty?

Yes, we were only able to evaluate the exact penalty by Monte Carlo.

Assigned_Reviewer_5:

> A graphical depiction of the spherical properties of the Fischer normalization described in lines 243-246 could be useful.

Thank you for this suggestion. We will add such an illustration to our next draft.

Assigned_Reviewer_6:

> Referring to Hinton's dropout in the introduction is a bit misleading as there is no models averaging in the current paper. It would be more precise to mention that this work consider another form of dropout, where we learn one model robust to Bernoulli multiplicative perturbation.

We will clarify our language to avoid any confusion.

> The non convexity of the regularizer poses computational issues. Authors mention in a footnote that they use lbfgs, more explanation and discussion are needed to understand how to avoid local minimas.

Questions surrounding the convexity of our method appear to be particularly interesting. Although our objective is not formally convex, we have not encountered any major difficulties in fitting it for datasets where n is reasonably large (say on the order of hundreds). When working with lbfgs, multiple restarts with random parameter values give almost identical results. The fact that we have never really had to struggle with local minimas suggests that there is something interesting going on here in terms of convexity. We are actively studying this topic, and hoping to gain some more clarity about it.

> The parameter delta/(1-delta) corresponds to a regularization parameter. any insight on how to choose delta? according to Hinton et al., delta=1/2 seems to have a good performance.

The tuning parameter delta/(1 - delta) behaves just like the lambda parameter in L2 regularization. We can set this parameter by cross-validation. In practice, we got good results by just using delta = 0.5 (i.e., delta/(1 - delta) = 1). We used delta = 0.5 in our experiments, but tuned delta for the simulation study.

> The log partition function is sometimes hard to get in closed from any insight on how to go around that?

Good point. The log-partition function is always tractable for the examples discussed in our paper, and this is part of what makes our method much faster than actual dropout. However, in some applications, the log-partition function can be more difficult to work with. Thankfully, we can often use special case tricks to do efficient computations with the log-partition function even when it does not allow a closed-form representation. We have not tried to apply our method to fully generic exponential families in which there is no way of efficiently getting the partition function.